# A Planning Model for Fire-Resilient Landscapes in Portugal Is Riddled with Fallacies: A Critical Review of "FIRELAN" by Magalhães et al., 2021

Nuno G. Guiomar [1,2,3], José M. C. Pereira [4] and Paulo M. Fernandes [5,*]

1 MED—Mediterranean Institute for Agriculture, Environment and Development & CHANGE—Global Change and Sustainability, University of Évora-PM, Apartado 94, 7006-554 Évora, Portugal; nunogui@uevora.pt
2 EaRSLab—Earth Remote Sensing Laboratory, University of Évora-CLV, Rua Romão Ramalho, 59, 7000-671 Évora, Portugal
3 IIFA—Institute for Advanced Studies and Research, University of Évora-PV, Largo Marquês de Marialva, Apartado 94, 7002-554 Évora, Portugal
4 Forest Research Centre, Associate Laboratory TERRA, School of Agriculture, University of Lisbon, Tapada da Ajuda, 1349-017 Lisboa, Portugal; jmcpereira@isa.ulisboa.pt
5 CITAB—Centro de Investigação e de Tecnologias Agro-Ambientais e Biológicas, Universidade de Trás-os-Montes e Alto Douro, 5001-801 Vila Real, Portugal
* Correspondence: pfern@utad.pt

**Abstract:** FIRELAN was developed as a model expected to foster the resilience to fire and sustainability of a landscape that is based on a number of premises about fire behaviour. We critically review FIRELAN and find that flawed ecological concepts and terminology are used, and that six fallacies are pervasive throughout the paper, namely begging the question regarding the effectiveness of land cover changes; the appeal to nature on the preference of native species over non-native species; confirmation bias on the flammability of native vs. non-native species; the oversimplification of fire behaviour drivers; questionable causation regarding the effect of land cover on fire hazard; and non-sequitur in respect to the flammability–resilience relationship. We conclude that FIRELAN overall lacks supporting scientific evidence, both theoretical and empirical, and would be unable to deliver adequate wildfire mitigation. Recommendations are given to guide the landscape-level process of planning and implementing wildfire impacts mitigation.

**Keywords:** fire behaviour; fire hazard; flammability; fire management; forest resilience; Portugal





## 1. Introduction

FIRELAN has been proposed as a conceptual, normative model to increase the resilience to fire and sustainability of a landscape and reduce the likelihood of large fires [1] (herein Magalhães et al.). Two major components set the theoretical framework: "Landscape Fire Resilience" and "Ecological Sustainability". Under "Landscape Fire Resilience", FIRELAN considers: (i) fire behaviour as influenced by topography; (ii) tree flammability, which is a fire behaviour driver; (iii) landscape discontinuities, and (iv) the wildland–urban interface. Those dimensions, however, express landscape resistance to fire spread and impact, rather than its ability to recover from wildfire disturbance (fire resilience). The "Ecological Sustainability" component comprises (i) the ecological network and (ii) the ecological land suitability. However, the application of a conceptual model, tool, or instrument created to fulfil specific objectives different from those defined in FIRELAN (e.g., the Ecological Network following the basis set by Forman [2] and the "land morphology concept" as established by Cunha et al. [3]) implies an adjustment to the new context.

Here we examine to what extent the FIRELAN assumptions are validated by current wildland fire science and discuss the main flaws in the authors' contribution in general and

regarding wildfire mitigation schemes in particular. This is critical before any attempts to apply their approach to the real world and in other geographical contexts.

The study is riddled with fallacies, which are arguments that seem to be better than they really are [4–7]. Six of these stand out: (i) assuming the conclusion, colloquially known as "begging the question", which occurs when the premise and conclusion are the same statement, although expressed in different words [4]; (ii) "appeal to nature" [8], which consists of claiming that something is good because it is (perceived as) natural, or bad because it is (perceived as) unnatural; (iii) "confirmation bias" that consists of only looking for evidence that supports what is already believed [9]; (iv) "oversimplification" [10], which ignores relevant complexities to make something appear simpler than it is; (v) "questionable cause" [11], occurring when a causal interpretation is defended based on limited evidence and no attempt is made to rule out alternative explanations; and (vi) "non-sequitur" [12], when the premises of an argument are not relevant for its conclusions. The remainder of this review will individually address the fallacies incurred by FIRELAN and then discuss the overall potential of the approach to mitigate wildfire spread and severity.

## 2. The Fallacies in FIRELAN

### 2.1. Begging the Question: Effectiveness of Proposed Land Cover Changes

The empirical analysis of fire data is absent from Magalhães et al. Indicators of the fire regime in mainland Portugal show high spatial variability, but the authors chose not to analyse the abundant fire data available on the number and location of ignitions; the extent and location of areas burned, fire causes, the timing of fires; and on the main land cover types affected [13–15], namely those obtainable from the Institute for Nature Conservation and Forests website (https://www.icnf.pt/florestas/gfr/gfrgestaoinformacao, accessed on 15 October 2022).

FIRELAN prescribes what Magalhães et al. think needs to be carried out to accomplish their stated goal, focusing on changes in land cover and in the relative abundance of different tree species. It is repeatedly affirmed that those land use and species composition changes will accomplish the goal of reducing fire incidence in their study area, but they fail to provide any evidence for the claim. Establishing the effectiveness of FIRELAN proposals would require either (i) a simulation analysis comparing the fire regime features of the current landscape and those of a landscape modified according to FIRELAN prescriptions, or (ii) a correlational study to test whether the FIRELAN-proposed changes in landscape structure and composition would accomplish the postulated changes in fire regime. The relevant fire regime, environmental, demographic, and socio-economic data available at the national level cover a broad range of landscapes and fire regimes that would allow for that kind of analysis.

In the absence of empirical analysis of fire data and of its environmental and socio-economic correlates, Magalhães et al. end up making a series of unsubstantiated statements, namely:

(i) *"This network ensures the effectiveness of discontinuities in the landscape with less combustible land uses. It also functions as a fire-retardant technique and protection of wildland-urban interface (WUI)."* (in the Abstract)—Results capable of substantiating this statement are not presented. This is wishful thinking.

(ii) *"The results show that land-use and tree species composition should change drastically, whereas about 72% of the case study needs transformation actions."* (in the Abstract)—Again, this is an assumption, in this case based on the misinterpretation of published research, and disregards research presenting contrary information.

(iii) *"This paper is significant for the Portuguese rural fires planning legal framework because FIRELAN plans, as the proposed land-use plan showed, can give explicit indications of adequate land-uses for fire prevention and sustainability. This is a very significant contribution to solve the problem."* (in the Discussion)—As in (i), this statement is unsupported by the research presented in the paper. Analytical results are not provided, based either on observed or simulated data, by comparing observed fire incidence patterns in the

current landscape with those that would occur in a landscape with the proposed land cover composition and structure.

(iv) *"Edges and swales with ponds, at a more detailed scale, are also useful to reduce the extension of rural fires. The results show that land-use and tree species diversity should change drastically (...)"* (in the Discussion)—There are no analytical results to substantiate the claim that land use and tree species diversity should change dramatically, because a simulation study comparing the fire regime under the current conditions and the proposed scenario was not carried out, nor was the contemporary observed fire regime in the study area compared with that of a region at another location, with a land use reasonably comparable with the FIRELAN model proposal.

Evidently, it is not possible for Magalhães et al. to ensure the effectiveness of less flammable land uses, namely their capability to slow down wildfire spread, which is highly dependent on the prevailing weather conditions; they present no quantitative results related to potential fire behaviour showing that tree species composition should change drastically; and the FIRELAN conceptual model lacks the capability to assess the adequacy of various land use types to prevent fire occurrence or mitigate fire behaviour. FIRELAN also lacks the analytical tools required to establish the need to create discontinuities in the landscape (e.g., [16,17]), or to assess the impact of structures such as edges, swales, and ponds on fire rate of spread, fireline intensity, or fire exposure (e.g., [18–20]). In each of these cases, assumptions about the fire-proneness of diverse land cover types are simply restated under the form of conclusions concerning their expected effects on fire behaviour. No evidence, theoretical or empirical, observed or simulated is provided to support conclusions, especially when quantitative.

The last sentence in the Discussion acknowledges the need for model validation based on fire behaviour simulation. The value of calibrating fire model outcomes is limited in the context of fire management planning. Nonetheless, it is a step that ought to have been performed prior to issuing numerous and detailed recommendations on drastic land use/land cover change aimed at fire prevention.

### 2.2. Appeal to Nature: Putative Advantages of Native vs. Non-Native Species

The appeal to nature fallacy infers goodness from naturalness, i.e., it claims that something is good because it is natural or bad because it is unnatural [8]. The appeal to nature fallacy pervades Magalhães et al., but the most evident statement occurs in Section 4.3 in [1], on the potential land-uses of FIRELAN: "The choice of native species referred to in the theoretical framework must be selected according to potential natural vegetation". Here is a clear value judgment on the goodness of native species, which leads to a prescription for their use, derived from the descriptive fact of their naturalness. Daston and Vidal [21] deserve to be quoted in full on this issue: "To invoke Nature in this way is to establish an allegedly disinterested view from nowhere, to pronounce a claim to scientific objectivity. The effacement of the one speaking on Nature's behalf is among the most powerful effects of the appeal to Nature".

Reasoning from an evolutionary perspective, Gould [22] questioned the notion of intrinsic superiority of native plants and stated that equating native with best adapted is fallacious in evolutionary terms because "natural selection... is not an optimizing device". He countered the functional argument based on adaptation in favour of native species by stressing that many native plants, adapted to their regions as a result of natural selection, fail competitively against newly introduced species, which lack evolutionary experience with the local habitat. Had natural selection yielded optimally adapted native species this could not happen because, by definition, the natives would outcompete all newcomer species [22].

The argument holding that the natural geographic ranges of plants reflect maximal ecological appropriateness can also be dismissed [22]. Contingent and random factors, namely those affecting plant dispersal and colonization, play such important roles in biogeography that natural selection cannot ensure that species occupy the geographic areas

most adequate to their biological features. For example, the ranges of many widespread forest species in Europe are estimated to be still limited by post-glacial migrational lag, and although species ranges are also strongly influenced by climate, it is likely that most forest tree species in the region will be unable to closely track the environmental changes induced by global warming [23].

Gould [22] also denounced what he designated "strict nativism", which considers any human intervention on the environment as intrinsically bad, denying our ability to make intelligent and ethical choices. Steinbock [8] offered a similar view when she defended that although nature does and should have value for us, it cannot be the source of substantive moral rules; on the contrary, it is a subject for moral assessment. A final point, especially relevant considering the goals of Magalhães et al. was raised by Kaebnick [24] and moves the topic from the domain of moral to political philosophy. He asked whether appeals to nature may be allowed to affect public policy, questioned the extent to which they may be used as "moral trump cards" or, on the contrary, need to be weighed against other moral concerns.

### 2.3. Selective Use of Facts: On the Flammability of Native vs. Non-Native Species

Confirmation bias, or selectively using facts, is the fallacy that consists of only looking for evidence that confirms what one wants to believe, or what one already believes [9]. This fallacy is particularly evident in Section 2.1.2 in [1], starting out with the assumption that native species are less flammable than non-native species, particularly *Pinus pinaster* (Pp) and *Eucalyptus globulus* (Eg) and only review studies that they believe confirm that assumption. Pp is a native species, as confirmed by several palaeoecological studies (e.g., [25]), with recent evidence showing that it never became extinct in Portugal [26,27]. However, Magalhães et al. contradict their own narrative–argumentative approach when considering the use of archaeophytes, exotic plant species that were introduced in "ancient" times.

Section 2.1.2 in [1] is very confusing, because the concept of tree species flammability is used so loosely that Magalhães et al. attempt to back their assumption by citing an ignition risk paper [28], which is a substantially different topic and irrelevant for tree flammability assessment. A sub-sample of the spatial distribution of ignition points included in the Spanish Forest Fire Statistics was used in [28] to assess the factors driving fire ignition risk in relation to vegetation type, topography, and wildland–urban interface. Moreover, the factors that drive the spatial distribution of ignitions are generally different from those that control burned area (e.g., [29]) and, therefore, such results should not be used to frame decision-making processes aimed at reducing burned area, fire size, or fire severity. Another paper cited [30] is about ecological succession influenced by differences in litter flammability in the Eastern USA; since all species involved are native, it is difficult to understand how it argues in favour of the lower flammability of native species.

Two other papers [31,32] assess the relative fire-proneness of various forest types, which Magalhães et al. also place under their loose interpretation of flammability. The utility and objectivity of the flammability concept itself has been questioned, e.g., [33], and many efforts are still needed to eliminate its main gaps, including the need to standardize methods, clarify terminology, and compare laboratory-based results to field assessments of fire behaviour [34]). Magalhães et al. inaccurately quote results on the fire-proneness ranking of forest types [31], stating that Eg plantations are more fire-prone than unspecified hardwood forests. In reality, Table III in [31] shows the results for fire-proneness assessed with three different approaches and their average value, which is virtually identical for Eg plantations and unspecified hardwood forests. Other hardwood species analysed in the study (chestnut, cork oak, and holm oak) are less fire-prone, but they are part of agroforestry systems that have substantially distinct vegetation structure and management, such that differences in fire-proneness cannot be exclusively assigned to tree species. A study on the fire proneness of various land cover types at European scale [32] is misrepresented as dealing with fire risk, and its most relevant conclusion for the topic of this section is that shrublands were the most fire-prone land cover.

Magalhães et al. also omitted [35] in this context, although later cited it to support the need to increase landscape heterogeneity. Moreira et al. [35] concluded that fire selectivity varies little among the more flammable forest types and shows high spatial variability. According to their spatial stratification, the selection ratios in the FIRELAN study area (see the results obtained for Beira Serra) showed a strong preference of fire for shrublands. Beira Serra was one of the regions where the fire preference for broadleaved forests was highest, being more preferred by fire than Eg and less preferred than conifers. Selection ratios obtained for Eg were similar to those calculated for agroforestry and silvopastoral systems with an evergreen oak tree layer.

The other side of confirmation bias is ignoring evidence against one's held beliefs [9]. This means disregarding research showing that native vegetation types are not less fire prone than non-native types, and research demonstrating that management is a more important determinant of fire-proneness than species identity. A study on fire selectivity towards various land cover types as a function of fire size [36] is equally neglected by Magalhães et al. Its most relevant conclusion for the topic of this section is that fires in mainland Portugal "prefer" Pp stands, but "avoid" Eg plantations slightly. Moreover, the study found that fire selectivity decreases with increasing fire size (see also [37]). The size of large fires in Portugal increases with landscape-level fuel connectivity and decreases with pyrodiversity, but these changes are independent of forest type [38]. A multi-approach assessment found no evidence to support the thesis that Eg afforestation altered the fire regime in Portugal [39]. Finally, [40] performed a detailed analysis of relative fire-proneness in Portugal over a long period of time (1975–2018) and for a large number of land cover types, using the likelihood ratio. They found chestnut forests (the archaeophyte *Castanea sativa*) to be the most fire-prone of all forest types analysed, followed by forests dominated by various native oak species (mostly *Quercus pyrenaica*, *Q. robur*, *Q. faginea*). Native Pp and exotic Eg plantations had slightly lower fire likelihood ratios but were clearly higher than those of riparian broadleaved and of short-needle conifer forests. Thus [40] does not corroborate the assumption of lower propensity of native species to burn defended by Magalhães et al.

The issue of fire selectivity for various forest types was approached by [41] from a broader perspective and included forest and understory structural attributes in the analysis. The combination of dominant tree species and variable stand structure yielded 19 forest types. The simulation of fire behaviour for those types revealed the primacy of stand structure over species identity as a driver of fire hazard, assessed by fire spread rate, fireline intensity, and likelihood of crown fire. While a simulation-based study has caveats, those findings were confirmed by a subsequent empirical analysis of variation in fire severity [42], which is a combined outcome of variation in fire behaviour and in stand structure. In this regard, Figure 1 is illustrative of the variation in fire severity that can be observed within a given forest type burned by the same wildfire.

Forest structure is determined by a number of factors, namely forest type, stand age, site characteristics, and management. To examine in greater depth how stand structure relates with fuel characteristics and potential fire behaviour, we revisited a fuel hazard database associated with the Portuguese national forest inventory plots [39]. This study had found that, depending on the surface fuel metric in question, forest type accounted for only 1.4–8.6% of the observed field variability. A K-means cluster analysis was carried out for each generic forest type (evergreen broadleaves, deciduous broadleaves, eucalypts, pines), excluding alien invaders, to classify stand structure as a function of tree basal area ($m^2\,ha^{-1}$), dominant height (m), and number of trees per hectare. For each forest type, four classes resulted from the analysis, denoted as 1 to 4 and representing increasing maturity. Then, we fitted generalized linear models to fuel metrics (log-transformed) with forest type, stand maturity class, and their interaction as independent variables, expecting that a relevant differentiation would emerge from the analysis. However, like in [39], the amount of explained variability was mostly very low, with $R^2$ values of 0.02, 0.03, 0.08, and 0.09, respectively, for understorey vegetation fuel loading, the percentage of dead fuel in shrubs,

total fuel loading, and fuel depth. Relevant, albeit moderate, levels of variability could be accounted for via the models for litter loading ($R^2 = 0.29$) and the percentage contribution of broadleaved shrubs to shrub loading ($R^2 = 0.31$). In Figure 2, we plot the outcomes of the best performing models ($R^2 > 0.05$) for the various combinations of forest type and stand maturity, where fuel hazard indicators are expressed on a 0–100 scale; the relative loading of broadleaved shrubs was inverted prior to the 0–100 scale conversion, as fuel hazard decreases with its increase.

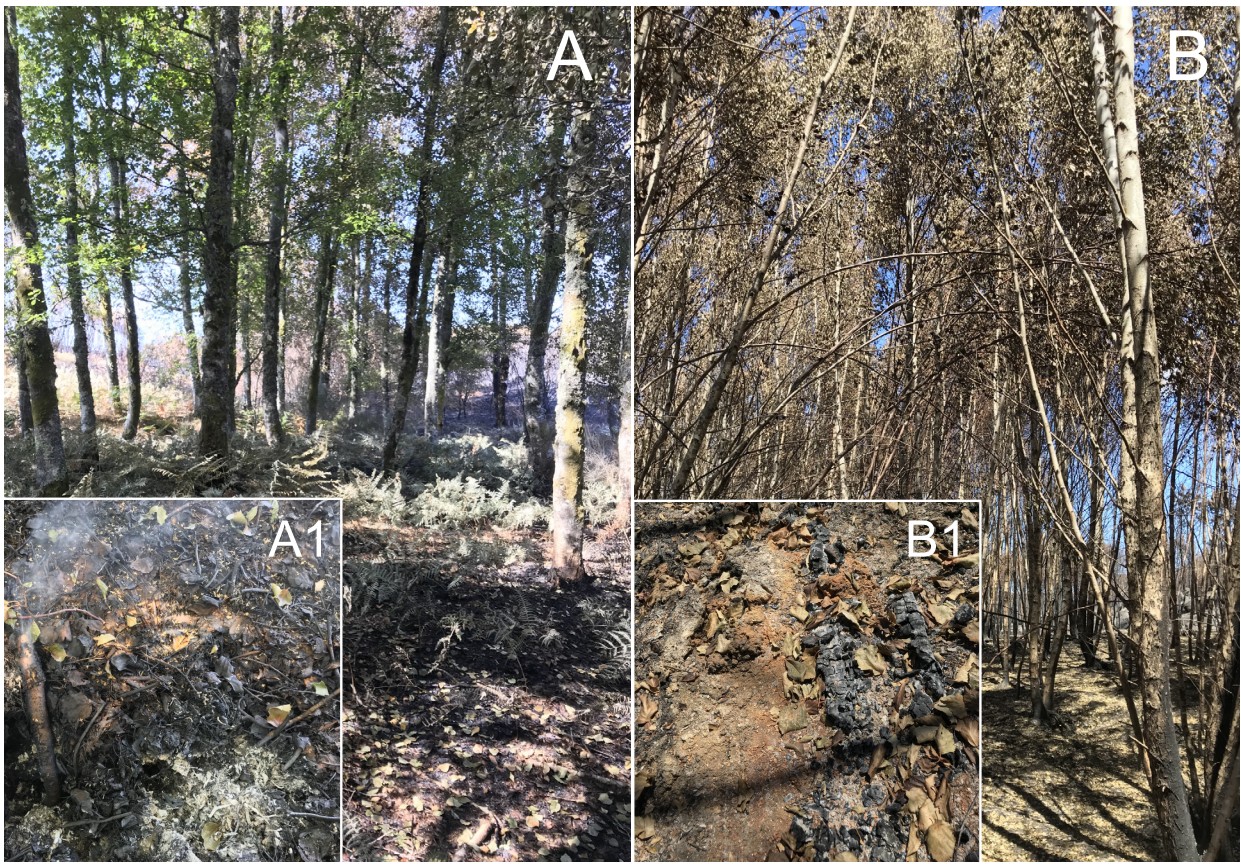

**Figure 1.** Fire severity in two downy birch (*Betula pubescens*) stands impacted by a 5557-ha wildfire in Serra do Alvão, Portugal, 21 August 2022. Fire danger was Very High (FWI = 37) on the day the fire started. Stand (**A**) and stand (**B**) were located in slopes facing east and west, respectively, and so fuels were potentially drier in stand A; in addition, A burned in the afternoon and B burned in the evening. Stands (**A**,**B**) were of similar height, but (**A**) was less dense, with higher canopy base height and was composed of larger diameter trees, while (**B**) displayed lower structural maturity and multistemmed individuals as a result of resprouting after a previous wildfire. Surface fuels in A comprised litter and a nearly continuous layer of ferns and grasses with scattered shrubs, but in B, only litter and downed dead woody fuels were present. The forest floor load was probably lower in stand (**A**) owing to a history of frequent low-intensity fire (four fires between 2000 and 2013 as indicated by the Portuguese fire atlas, https://geocatalogo.icnf.pt/catalogo_tema5.html, accessed on 6 September 2022). Stand (**A**) was burned by the right flank of the fire and stand (**B**) was burned by the head fire, i.e., by the faster spreading and more intense section of the fire front. Differences in fire severity are manifest and express the compounded effects of those influences: in (**A**), patchy burning, the forest floor partially consumed (**A1**), the live understorey mostly unburned, very low bole char height, and the canopy mostly unscorched; in (**B**), deep forest floor consumption implying high burn severity- as revealed by the reddish colour of the soil (**B1**), higher stem charring, and total crown scorch, including some degree of foliar charring and combustion at the bottom of the canopy. Photos taken within one week after the fire by P.M. Fernandes.

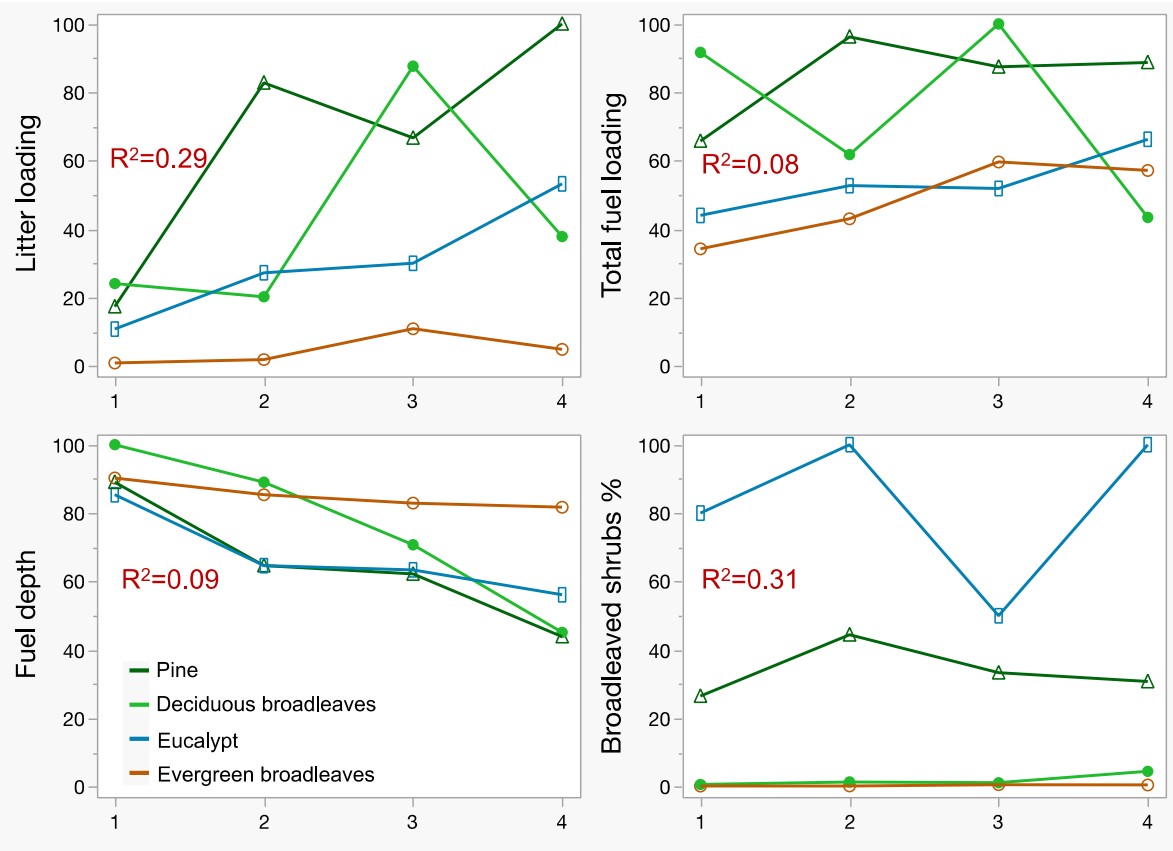

**Figure 2.** Descriptors of surface fuel hazard in the National Forest Inventory plots (*n* = 3271) modelled as a function of forest type and stand maturity (increasing from 1 to 4). Higher values denote higher fuel hazard. The relevant species in the dataset per forest type were evergreen broadleaves—*Quercus suber* (18.3% of the total number of plots), *Q. rotundifolia* (15.6%); deciduous broadleaves—*Quercus pyrenaica* (3.5%), *Q. robur* (0.9%), *Q. faginea* (0.2%), *Castanea sativa* (0.6%); eucalypts—*Eucalyptus globulus* (23.1%); and pines—*Pinus pinaster* (33.1%), *P. pinea* (2.8%), *P. sylvestris* (0.3%), *P. halepensis* (0.2%).

Figure 2 indicates variable fuel dynamics, depending on forest type and fuel hazard component. Fuel hazard changes are negligible in evergreen broadleaves, probably because most stands are open and are often heavily managed and grazed. The most consistent temporal trend is for fuel depth, which decreases as structural maturity is acquired. Fuel loadings are distinctively higher in pine stands and intermediate in deciduous broadleaves and eucalypt stands. The most distinctive feature of native broadleaves is a substantially higher contribution of broadleaved shrubs (mostly the overstorey species regeneration) to shrub fuel loading, regardless of stand structure; nonetheless, and contrary to expectations, the lowest relative abundance of broadleaved shrubs in deciduous stands is found in highest stand maturity class. Is fire behaviour potential in native broadleaved forest lessened by the reduced understorey flammability? The answer is no, because the fireline intensity simulated for the forest inventory plots by [39] barely decreases with the percentage of broadleaf shrub loading, with $R^2$ values of 0.02 and 0.05 for evergreen and deciduous broadleaves, respectively. Therefore, the quantitative analysis of the relevant empirical data does not support the FIRELAN assertion that fuel hazard in native broadleaved forests decreases with stand age and maturity.

It is reasonable to conclude that there is no generalizable evidence of intrinsically lower fire proneness of native broadleaved forests in relation to eucalypt and pine forests, even in mature stands. Fire-related traits can be observed both in native and in non-native species, and the dominant native vegetation types in Portugal are fire-prone, with understorey and litter fine fuels playing a major role in fire behaviour [41].

*2.4. Oversimplification: Drivers of Fire Behaviour*

Section 2.1.1 in Magalhães et al. [1] is about fire behaviour but oddly discusses it only in terms of its relationship with topographic features, namely slope, aspect, and topographic position. Vegetation, weather, and their relations with topography are not addressed in this section, resulting in a grossly oversimplified view of fire behaviour. The four specific statements that make up most of Section 2.1.1 in [1] are also fallacious oversimplifications, since they attempt to make the issue appear simpler than it really is by ignoring several relevant complexities [10] and overemphasizing the role of one factor [4].

Topography has complex effects on fire behaviour and, over time, on the fire regime (e.g., [43,44]), and interacts strongly with weather and vegetation [45]. Instead, FIRELAN offers (as rules of thumb) four bullet point sentences that are inaccurate at best:

(i)　"North aspect hillslopes, with a slope higher than 25%, by receiving less radiation throughout the year, burn less than the other hillslope aspects". The lower fire likelihood of Northern aspects and slopes > 25%, attributed to a decrease in solar radiation exposure, is based on a Mediterranean Europe-scale study [46]. However, the study shows that the effect size of aspect is very low and is irrelevant from the standpoint of fire planning. Conversely, [47] found higher fire selectivity for northern and eastern-exposed slopes in northern Portugal but, again, the difference was irrelevant for practical purposes. Figure 3, for two sample areas in Portugal as an illustrative example, shows that fire frequency (or burn probability) is very similar between aspect classes. Fire preference for a given aspect class and the resulting landscape-level patterns can be a mere outcome of wind-topography direction alignments, sometimes visible at regional scales [48]. While steeper terrain is usually associated with increased fire activity [47], more complex and dissected terrain can locally restrain fire spread and size [38], because of the correlation with fuel discontinuity and modified wind and fuel moisture patterns [49,50].

(ii)　"The fire progression speed doubles for every 10° (about 17%) increase in slope, and it can rise continuously in steep hillslopes from bottom to ridge". This rule of thumb for fire spread rate is well known and is an outcome of the existing empirical models [51–53]. However, it is unknown whether the rule extends to slopes > 30° due to insufficient experimentation, confounding effects in the field, and lack of a fundamental understanding of the effect of steep slope on fire behaviour [54].

(iii)　"Above slopes higher than 30° (57%), the relationship between the slope and fire speed is almost exponential". The exponential effect of slope on rate of fire spread applies over the entire variation of positive slopes in all existing models; again, information for steeper slopes is quite scarce [54].

(iv)　"When the fire reaches the top of the river basin (ridge) if it does not progress to the opposite side due to the hillside breeze, it begins to plough along the contour lines losing speed". This assumption is valid only for calm conditions when the wind is topographically (convectively) induced, and even this is not valid beyond mid-afternoon when the direction of topographic winds is reversed because of differential heating effects [55]. While wind-driven fires spreading downslope will experience some decrease in rate of spread [54], acceleration can also occur because of the well-known Foehn effect [56]. In fact, it has long been known [57] that blow-up fires can override topography. Extended droughts and extreme fire behaviour phenomena, e.g., ridge to ridge or slope to slope spotting can override topographic effects on fire spread, and landscape-scale wildfires spread at an average rate similar to that observed in flat terrain ([54] and references therein).

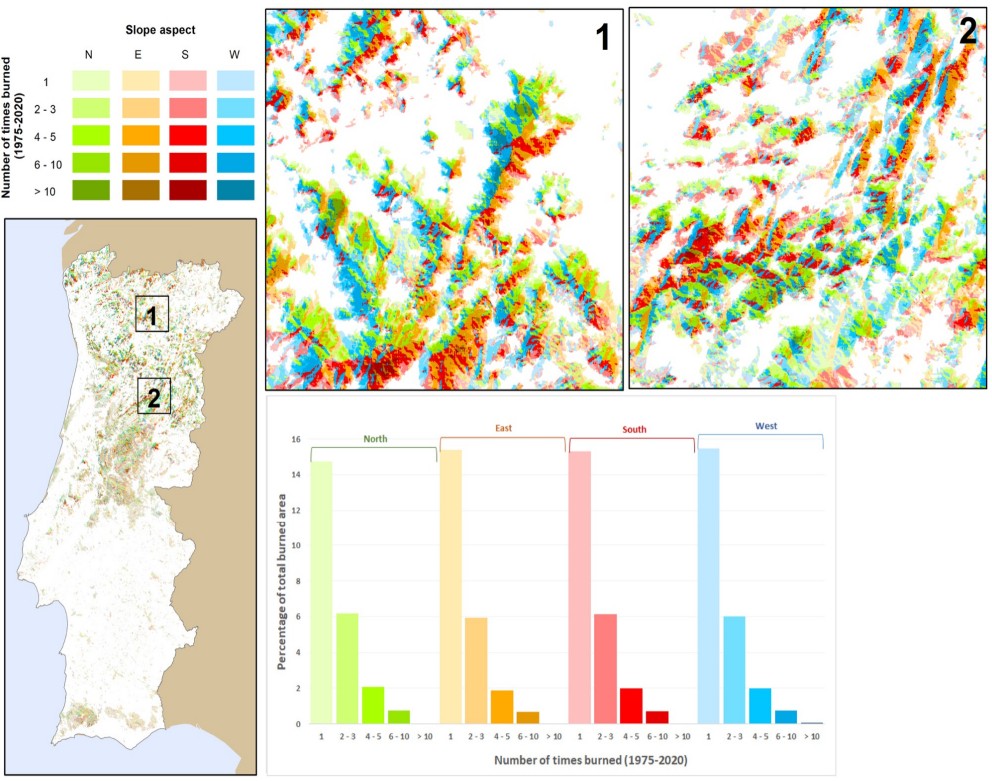

**Figure 3.** Fire recurrence (1975–2020) by slope aspect class for two example regions in Portugal. In region 1, fire regimes 3C (some large fires, but low frequency of occurrences) and 4B (short fire season and low burned area, large fires are absent) prevail; while in region 2, fire regimes 1B (high and regularly burned area), 3B (mega-fires and large burned area), and 3C stand out (classification of fire regimes at the parish scale by [58]). Despite the fire regimes' heterogeneity in these two regions, the distribution of fire recurrence is very similar between the different slope aspect classes.

*2.5. Questionable Cause: Land Cover and Fire Hazard*

　　The questionable cause [59] arises when (i) there is a causal claim in the argument, (ii) the causal claim is championed without the proper support for it, and (iii) there is reason to question the acceptability of the claim. The main issue in this fallacy, also called objectionable cause [11], lies in the way the conclusion is hastily reached, without ruling out alternative explanations.

　　In the Introduction, Magalhães et al. state that (i) "(. . .) the North region displays most fire occurrences and concomitantly the Eg and Pp are the dominant species [22]. Therefore, this indicates that land-use and land cover are major factors in fire risk increase" and (ii) "(. . .) in the southern region, whenever Eg is planted intensively, fires in those areas become more intense and disastrous, as happened in Serra de Monchique in the Algarve region with an estimated burnt area of around 28,000 ha by the 2018 fire". These two statements are a case of questionable cause fallacy because the fact that two events occur together is taken to imply a cause-and-effect relationship. In the first statement, the authors disregard the fact that the vast majority of ignitions in the northern part of the country occur in densely settled, predominantly urban and agricultural areas (see Figure 2 in [60]; Figure 4 in [61]; Table 3 in [39]; and the spatial distribution of fire regimes in mainland Portugal in [58]). Therefore, the number of ignitions is not an outcome of the extent of area occupied by Pp and Eg stands. Regarding the second statement, while large fires might have become more frequent after the extensive afforestation of Serra de Monchique, the 2018 fire burned Eg plantations and cork oak woodlands in nearly identical proportions to their presence in the area [62], i.e., the fire neither preferred nor avoided either land cover type, according to the terminology of fire selectivity studies. The same report found a correspondence between

higher fire growth rates and higher proportion of shrubland and cork oak woodland in the landscape. This indicates that factors other than tree species determined fire spread and extent. In addition, a very large fire occurred in Serra de Monchique in the mid-1960s, prior to Eg afforestation, that burned shrublands, oak woodlands, strawberry tree stands, and pine stands [63]. Thus, the extent to which Eg afforestation can be blamed for large fires in the region, namely the 2018 event, is questionable. Additionally, the assumption that fire impacts are proportional to fire size is inaccurate [64] or plain wrong, e.g., [65].

### 2.6. Non-Sequitur: Combustibility/Flammability and Resilience

FIRELAN "(...) establishes the need to create discontinuities in the landscape with less combustible land uses and apply permaculture techniques, such as swales and ponds, towards a more fire resilient landscape". ([1], p. 21). Combustibility is a somewhat vague concept but is one of the dimensions of flammability, and it was suggested [66], and subsequently adopted in the flammability literature, to be the rate of weight loss during combustion. Thus, combustibility is inadequately used as a synonym for flammability. Also, land uses are not flammable, even if flammability can be indirectly influenced by land use, because flammability is a property of vegetation or land cover. The non-sequitur fallacy occurs when the premises of an argument are not relevant for its conclusions [12]. We believe this to be the case for the statement above, as detailed below.

Ecological resilience can be defined either by "the amount of disturbance that an ecosystem could withstand without changing self-organized processes and structures" or by the "return time to a stable state following a perturbation" [67]. Fire has been playing a critical natural selection role in the evolution of terrestrial plants (e.g., [68]). Several fire-adapted and fire-related traits can be observed in fire-prone ecosystems, resulting from a fire-influenced evolutionary process (e.g., [69,70]) and comprising both fire-survival and fire-embracing strategies [71]. Therefore, lower resilience may be associated with vegetation with limited evolutionary exposure to fire, and thus lacking the adaptations required to ensure the ability to recover repeatedly, extensively, and quickly after burning. For example, *Abies alba* and *Fagus sylvatica* are species typical of low flammability forests, but the former is fire-intolerant and the latter is fire-sensitive [72], in both cases due to thin bark [73]. The highly flammable shrubland communities in the Mediterranean basin are highly resilient to fire, especially when resprouter species are well represented [74]. Post-fire tree responses are mainly determined by their persistence strategy, i.e., individual survival through passive resistance or resprouting vs. population persistence through germination and how these strategies interact with fire severity (e.g., [75–77]) but also by the cumulative effect of fire and other disturbances (drought, pests and diseases, land management) (e.g., [78–80]). Magalhães et al. oddly chose to neglect the extensive literature addressing fire ecology as well as tree mortality and post-fire regeneration in the Iberian Peninsula in relation to different dimensions of fire regimes [75–78,81–91]).

Thus, lower flammability does not necessarily lead to higher resilience, hence the *non-sequitur* fallacy. FIRELAN proponents might have had in mind socio-economic resilience, rather than ecological resilience, but in that case the low flammability of the vegetation making up the proposed land cover does not necessarily imply that it will provide the social and economic sustainability required for resilient socio-ecological systems, leading to another instance of non-sequitur.

## 3. Could the FIRELAN Strategy Achieve Landscape-Level Wildfire Mitigation?

A fundamental issue in the FIRELAN approach is how landscape fire-spread mitigation strategies are boiled down to either forest type conversion and farmland expansion, a difficult endeavour with no guarantees of success, as previously shown, or to isolation through linear discontinuities. Area-wide fuel reduction treatments that can potentially disrupt fire spread and decrease wildfire size and effects (e.g., [92–95]) are not an option in FIRELAN.

The limitations of the isolation strategy in compartmentalizing fire spread are well known, stemming mostly from wildfire characteristics (size, intensity, orientation, spotting potential) and whether the fuel breaks are actively used by firefighting resources during wildfire containment operations (e.g., [96–98]). Figure 4 provides a recent example from Portugal. A high percentage of fires are expected to burn over the most important fuel breaks in Portugal, allowing us to conclude that it is critical to enhance landscape-level fuel reduction to achieve short-term fire management objectives (reduce large and high-severity wildfires) [17]. The use of linear fuel breaks is recommended to anchor area-wide fuel reduction treatments, such as prescribed fire [92]. The passive effect of these linear fuel breaks is very low (e.g., [98]), and their effectiveness increases dramatically if used in fire suppression activities [97]. FIRELAN is partial to green belts (or green strips), a fuel break variant where fuel reduction is replaced by conversion to a low-flammability fuel type. While green fuel breaks have their place in fuel management, especially for protection purposes in the rural–urban interface [99] their limitations in relation to wildfire containment are the same as those of fuel-reduction breaks and the existing evidence of their effectiveness is anecdotal [100].

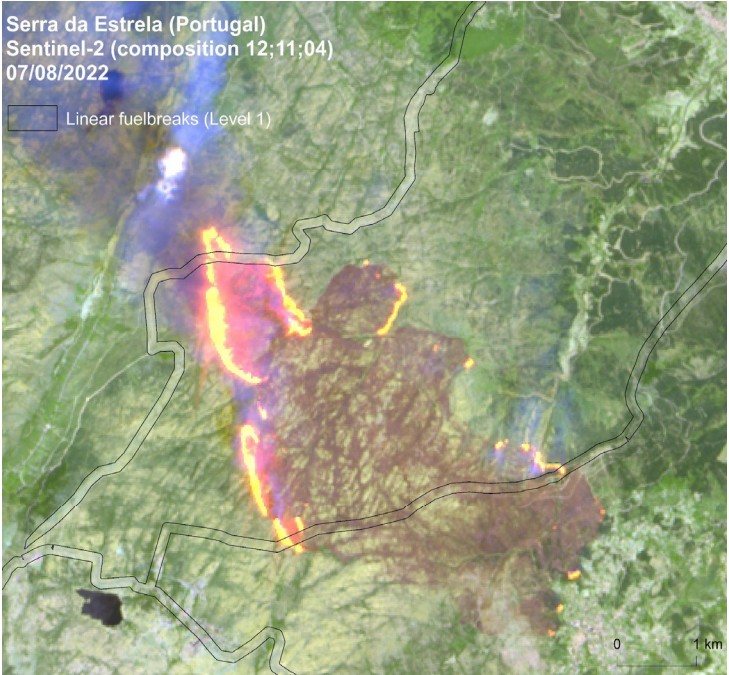

**Figure 4.** Linear fuel breaks (approximately 125 m wide) crossed by a large wildfire that occurred in August 2022 in Serra da Estrela, central Portugal.

As the effectiveness of fire suppression is limited by fire behaviour, the complementarity between the strategies highlighted by Agee et al. [92] would substantially increase the probability of firefighting success. In fact, small-scale fuel reduction treatments scattered across the landscape are ineffective in reducing fire growth and damage. Prescribed burning in Portugal has a modest effect on subsequent wildfire extent, as treatment effort on the landscape is low and most treatment units are small enough to be overrun by large fires [101]. To cope with the current fire regime and landscape changes (particularly those reflecting land abandonment), future fuel-treated patches must be substantially larger, and should be planned in conjunction with existing or proposed land cover and linear fuel breaks. It is critical to evolve to a fire-adapted silviculture to deal with the accumulated fuels across rural landscapes, including prescribed burning in forests [102].

Ignoring the role of fuel-reduction treatments is part of FIRELAN's assumption that fire behaviour is modified solely by changing forest composition. As indicated by basic fire behaviour science and demonstrated by the previously mentioned fire preference

studies and burn probability likelihoods, such premise ignores that forest structure and fuel accumulation often override forest composition (e.g., [41]). Lower fire severity has been observed in native deciduous and evergreen broadleaved forest than in adjacent pine forest, but the change in forest type explained less of the existing variation in fire severity than variation in forest structure, topography, and fire spread patterns [103]. Hence a change in forest composition requires follow-up through silviculture and fuel treatments, at least for a certain amount of time (up to a few decades). Pais et al. [103] used process-based fire regime modelling to examine the outcomes of alternative land use/land cover scenarios in a transboundary region of NW Iberia. Magalhães et al. postulate the fire-hazard reduction benefits of conifer forest conversion to oak woodland, and yet such a strategy would reduce wildfire area only when combined with farmland expansion [103].

Finally, FIRELAN proposes interventions and techniques such as swales and mulching that are expected to be particularly ineffective for wildfire spread mitigation, besides being impractical beyond agricultural soils. Moist or even flooded soils do not hamper fire spread and fire behaviour in marsh swales has been described as "spectacular" [104].

## 4. Conclusions and Recommendations

The landscape envisioned under the framework of FIRELAN offers no guarantee of reduced vulnerability, nor of increased resilience to fire, contrary to the claims of its proponents, because it was not confronted with any fire ignition or burned area data, neither observed nor simulated. The entire process of geoprocessing and map algebra was based on a theoretical framework containing flawed ecological principles, loosely used scientific concepts, and numerous fallacies. Moreover, scientific findings were cherry-picked or misrepresented to conform to a preconceived narrative, instead of relying on current knowledge to build or model the rules for the fire-hazard reduction proposal.

Reliance on assumptions, even if unrealistic or anecdotal, is not a concern if used to build and test hypotheses. Partial or contextual realities cannot be universalized, and Magalhães et al. lacked the basic step of validating their starting assumptions. The hypothetical landscape defined from their assumptions should have been tested and compared with a reference landscape, through stochastic wildfire simulation to assess the effect of different landscape-level fuel treatments, or by analysing land cover changes, e.g., [18].

The many flaws underlying FIRELAN are concerning beyond the academic realm. This conceptual model underlies the 55,000-ha Landscape Design and Management Programme (*Programa de Reordenamento e Gestão da Paisagem*, PRGP, in Portuguese) of Lousã and Açor for two mountainous regions of central Portugal. The governmental decision that established the Landscape Transformation Program (Official Journal of the Republic, 2020) states that the objective for PRGPs is "...to promote landscape design as the framework for a new economy in rural lands, promoting a multifunctional, biodiverse, resilient, and more profitable forest, with improved carbon sequestration capacity and ability to provide better ecosystem services". These are laudable goals, which we share. However, we do not believe that a conceptual framework as deeply flawed as FIRELAN will contribute to bring them about.

The guiding principles of the Landscape Transformation Program (*Programa de Transformação da Paisagem*, PTP, in Portuguese), which frames the PRGPs, reflect a sound understanding of the environmental, social, and economic dimensions of rural fires in Portugal. They place due emphasis on key aspects that must underlie the PTP's ambitious goals. These include requiring the implementation of "[...] a locally based participatory process that strengthens local culture and the capacity of local actors"; the adoption of public environmental policies that "[...] align the interests of society and future generations with those of landowners and managers to promote greater interregional and intergenerational justice, ensuring the proper valuation of rural property and the promotion of sustainable management"; the requirement for the application to rustic property of "[...] sustainable management as a pillar of rural land planning, making it viable in smallholding territo-

ries through its productive valuation and the recognition and compensation of positive externalities"; and the proposal to have a governmental agency undertaking the management of properties with no known owner. Government agencies leading the process of PRGP development and implementation must establish the proper mechanisms to ensure that these land management plans comply with the principles set by the PTP and are scientifically sound.

Changes in the composition of fire-prone landscapes require substantial investment, time, the testing of cost-effective management models, and the high involvement of actors with different visions of the territory. Regardless of their effectiveness, substantial landscape changes are unlikely in the current socio-ecological context and within the fire return interval of large and/or extreme wildfires. PRGP development should be adapted to complex problems to enhance proactive and sustainable fire management in the context of complex socio-ecological changes in marginal productivity lands. This general objective must consider landscape as a whole, and can be achieved by:

1. Implementing landscape strategy-making processes [105] involving all type of stakeholders to develop alternatives suiting the socio-ecological context, as landscape change scenarios cannot be imposed and must be discussed and built locally from the beginning of the planning process.
2. Understanding of the drivers of extreme fire behaviour and the diversity of fire regimes within the PRGP areas.
3. Planning and decision making guided by land management goals, acknowledging the inherent interconnectedness of human (well-being) and natural systems (carrying capacity) and their role in shaping the fire regime.
4. Establishing a fire management framework to reduce fire-induced damages and losses rather than the size of the burned area [64].
5. Promoting payment schemes targeting the integration of fuel reduction as a service to be paid by beneficiaries distributed at different levels, from the society to the neighbouring landowners of the intervened-in areas.
6. Prioritizing actions that are feasible within the time horizon of the PRGP, promoting ongoing and previously evaluated initiatives that can be upscaled.

**Author Contributions:** Conceptualization, N.G.G., J.M.C.P. and P.M.F.; formal analysis, N.G.G. and P.M.F.; investigation, N.G.G., J.M.C.P. and P.M.F.; writing—original draft preparation, N.G.G., J.M.C.P. and P.M.F.; writing—review and editing, N.G.G., J.M.C.P. and P.M.F. All authors have read and agreed to the published version of the manuscript.

**Funding:** N.G. was funded by the European Union through the European Regional Development Fund in the framework of the Interreg V-A Spain–Portugal program (POCTEP) under the CILIFO (Ref. 0753_CILIFO_5_E) and FIREPOCTEP (Ref. 0756_FIREPOCTEP_6_E) projects and by National Funds through FCT under the Project UIDB/05183/2020. J.M.C.P. was funded by the Portuguese Foundation for Science and Technology (FCT) under Project UIDB/00239/2020. P.M.F. was funded by the Portuguese Foundation for Science and Technology (FCT) under Project UIDB/04033/2020.

**Conflicts of Interest:** The authors declare no conflict of interest.

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
