# Peer review of "A Planning Model for Fire-Resilient Landscapes in Portugal Is Riddled with Fallacies: A Critical Review of “FIRELAN” by Magalhães et al., 2021"

_fire, doi:10.3390/fire6100398_

Round 1
Reviewer 1 Report
Review of A Planning Model for Fire-Resilient Landscapes in Portugal is Riddled with Fallacies: A Critical Review of “FIRELAN” by Magalhães et al., 2021
The authors critique the previously published FIRELAN conceptual plan to increase fire resilience and reduce the likelihood of large fires. It is a welcome and long overdue dissection of policies akin to dogma that have been repeated without examination against logic or physical science.
p. 125-126 One might also note that "calibration" (fudging) of fire models has become so pervasive, and even encouraged in fire science, that the outcomes of models - whether or not a particular behavior can be expected - are basically useless as guidance.
A related point for many of the arguments presented (perhaps especially the one beginning L117) is that treatment or species-dependent fire behavior that appears to have an effect in small prescribed fires in marginal burning conditions might not have a significant impact in large wildfires where environmental conditions are more supportive of fire growth - e.g. differences in fire spread in native vs. non-native grasses may appear while watching prescribed fires in marginal conditions but in severe fire conditions, the fire will spread rapidly anyway. In the latter, any mitigation may have limited effects.
L449-451. An equally critical view of whether or not these treatments can be logically concluded to have caused resultant fire behavior would be welcome. I'm not sure methods unequivocably support that conclusion; certainly most arguments selectively point at a treatment area and a fire that behaved favorably to the pointer's belief.
L457 - "Allowing us to conclude"
L472-482 If I've read it correctly, a summary of the authors' point is that treatments can be effective as landscape-scale mitigation, but not small-scale treatments, yet treatments need not be forest/land type conversion or linear fire breaks.
L504-506. Nice!
Author Response
The authors critique the previously published FIRELAN conceptual plan to increase fire resilience and reduce the likelihood of large fires. It is a welcome and long overdue dissection of policies akin to dogma that have been repeated without examination against logic or physical science.
Reply: Thank you!
125-126 One might also note that "calibration" (fudging) of fire models has become so pervasive, and even encouraged in fire science, that the outcomes of models - whether or not a particular behavior can be expected - are basically useless as guidance.
Reply: We introduced a sentence addressing this remark.
A related point for many of the arguments presented (perhaps especially the one beginning L117) is that treatment or species-dependent fire behavior that appears to have an effect in small prescribed fires in marginal burning conditions might not have a significant impact in large wildfires where environmental conditions are more supportive of fire growth - e.g. differences in fire spread in native vs. non-native grasses may appear while watching prescribed fires in marginal conditions but in severe fire conditions, the fire will spread rapidly anyway. In the latter, any mitigation may have limited effects.
Reply: The initial sentence of the paragraph was expanded and this point was added.
L449-451. An equally critical view of whether or not these treatments can be logically concluded to have caused resultant fire behavior would be welcome. I'm not sure methods unequivocably support that conclusion; certainly most arguments selectively point at a treatment area and a fire that behaved favorably to the pointer's belief.
Reply: We toned down the sentence statement to accommodate this concern. As fuel treatments are not considered in FIRELAN we don’t think further development is warranted.
L457 - "Allowing us to conclude"
Reply: Modified.
L472-482 If I've read it correctly, a summary of the authors' point is that treatments can be effective as landscape-scale mitigation, but not small-scale treatments, yet treatments need not be forest/land type conversion or linear fire breaks.
Reply: Yes. We edited the sentence end to make it clearer.
L504-506. Nice!
Reply: Thanks.
Reviewer 2 Report
The manuscript “A planning model for fire-resilient landscapes in Portugal is riddled with fallacies: A critical review of “FIRELAN” by Magalhães et al. (2021)” submitted to the journal Fire describes potential flaws with the FIRELAN model to increase resiliency on the landscape in Portugal. I feel that this paper is overall well written and important because. However, I have some few suggestions on ways to improve the manuscript (please see below):
1) Your criticisms of them using natural systems are justified. However, this blanket criticism could be blunted a little. For example putting statements like “mostly erroneous” (line 121), “loosely used buzzwords” (line 506) are not needed. These make the problems in the article seem personal.
2) If the government of Portugal is implementing aspects of FIRELAN are there a series of steps (bullets) that you can suggest to "fix" these flaws (i.e., proactive solutions)? This would at least provide information to the government that could be considered.
3) Below are some additional line x line comments.
Line X Line comments:
Figures 1-3 not visible or available on the website for manuscript.
449: Rephrase. A model doesn’t forget.
Author Response
The manuscript “A planning model for fire-resilient landscapes in Portugal is riddled with fallacies: A critical review of “FIRELAN” by Magalhães et al. (2021)” submitted to the journal Fire describes potential flaws with the FIRELAN model to increase resiliency on the landscape in Portugal. I feel that this paper is overall well written and important because. However, I have some few suggestions on ways to improve the manuscript (please see below):
1) Your criticisms of them using natural systems are justified. However, this blanket criticism could be blunted a little. For example putting statements like “mostly erroneous” (line 121), “loosely used buzzwords” (line 506) are not needed. These make the problems in the article seem personal.
Reply: We eliminated those statements.
2) If the government of Portugal is implementing aspects of FIRELAN are there a series of steps (bullets) that you can suggest to "fix" these flaws (i.e., proactive solutions)? This would at least provide information to the government that could be considered.
Reply: Thanks for the comment. We added a few sentences addressing this that now end the paper.
3) Below are some additional line x line comments.
Line X Line comments:
Figures 1-3 not visible or available on the website for manuscript.
Reply: The initial submission “lost” those figures, which were subsequently added, but it seems the corrected file was not made available to the reviewer.
449: Rephrase. A model doesn’t forget.
Reply: Done.
Reviewer 3 Report
This manuscript makes several important points in regard to the general application of any model to policy decisions.
Author Response
Reviewer: This manuscript makes several important points in regard to the general application of any model to policy decisions.
Reply: We thank the reviewer for the positive assessment.